# Programmed Grooming after 30 Years of Study: A Review of Evidence and Future Prospects

**DOI:** 10.3390/ani14091266

**Published:** 2024-04-23

**Authors:** Michael S. Mooring

**Affiliations:** 1Department of Biology, Point Loma Nazarene University, San Diego, CA 92106, USA; mmooring@pointloma.edu; 2Quetzal Education and Research Center, San Gerardo de Dota 11911, Costa Rica

**Keywords:** programmed grooming, stimulus-driven grooming, body size principle, vigilance principle, habitat principle, tick challenge principle, ticks, ungulates, rodents, ectoparasites

## Abstract

**Simple Summary:**

The programmed grooming model is an evolutionary hypothesis for the regulation of ectoparasite-defense grooming behavior. It proposes that grooming to remove ectoparasites such as ticks is regulated by a type of internal biological clock that has evolved to pre-emptively remove parasites before they can blood feed. Programmed grooming contrasts with the stimulus-driven model, in which grooming is stimulated by direct peripheral irritation from ectoparasites. Since programmed grooming was first proposed in 1992, 26 studies have provided robust support for the model with ungulate hosts and ticks. Additional studies from unaffiliated investigators have evaluated the predictions of the model in different host systems (including rodents and primates) and in a variety of ectoparasites (fleas, lice, and keds). After reviewing the current evidence, I conclude that (1) tests of the programmed grooming model should utilize the established protocol, so that the results can be compared and assessed in light of previous studies; (2) the model predictions should be tailored to the host biology under investigation; and (3) the model predictions should be tailored to the ectoparasite biology, since the efficacy of grooming depends on the parasite. It is hoped that future studies will reveal much more about how grooming helps wild animals to defend themselves against the threat of parasites.

**Abstract:**

In 1992, an evolutionary model for the endogenous regulation of parasite-defense grooming was first proposed for African antelope by Ben and Lynette Hart. Known as the programmed grooming model, it hypothesized that a central control mechanism periodically evokes grooming so as to remove ectoparasites before they blood feed. The programmed grooming model contrasts with a stimulus-driven mechanism, in which grooming is stimulated by direct peripheral irritation from ectoparasite bites. In the 30+ years since the seminal 1992 paper, 26 studies have provided robust support for the programmed grooming model in ungulate hosts and ticks. In addition, multiple studies from unaffiliated investigators have evaluated the predictions of the model in different host systems (including rodents and primates) and in a variety of other ectoparasites (fleas, lice, and keds). I conducted a tricennial review of these studies to assess the current evidence and arrived at the following three conclusions: (1) tests of the programmed grooming predictions should use a similar methodology to the well-established protocol, so that the results are comparable and can be properly assessed; (2) the predictions used to test the model should be tailored to the biology of the host taxa under investigation; and (3) the predictions should likewise be tailored to the biology of the ectoparasites involved, bearing in mind that grooming has varying degrees of effectiveness, depending on the parasite. Further research is warranted to enhance our understanding of the role of grooming in maintaining the health of wild animals in the face of parasite attacks.

## 1. Introduction

Grooming, broadly speaking, involves all forms of body surface care and is an important activity for the survival and wellbeing of animals. Whether directed to an individual’s own body (self-grooming) or to that of a conspecific (allogrooming), grooming is virtually ubiquitous among terrestrial vertebrates (Figure 1, Figure 2 and Figure 3). Of the many possible functions of grooming, parasite removal is likely to be the most important. The cost of ectoparasites for host animals has been well documented in animal production studies, including tick-associated declines in growth for domestic calves [1,2,3]. For example, a moderate tick load on a calf can result in a 10–44 kg reduction in weight gain per year, due to blood loss and tick-induced anorexia [4]. A similar loss of reserves in a wild animal would clearly have fitness-compromising consequences. The efficacy of grooming in removing ectoparasites has been established through experimental studies, in which grooming was restricted [5,6,7,8]. For example, impala (*Aepyceros melampus*), wearing a neck harness that partially prevented oral self-grooming, harbored 20 times more adult female ticks compared with impala wearing control harnesses that permitted grooming [9]. Given the significant costs of ectoparasite infestation, individuals exhibiting effective grooming behaviors would be at a selective advantage. However, grooming behavior itself is not without costs, including compromised vigilance against predators [10,11,12] and conspecifics [13,14], saliva loss from oral grooming [15], attrition of dental elements used for oral grooming [16], and the thermoregulatory costs of winter hair loss from excessive grooming in cold environments [17]. Thus, natural selection should favor an optimal grooming rate that correctly balances the cost of ectoparasite infestation against the costs of grooming.

In 1992, an evolutionary model for the endogenous regulation of tick-removal grooming was first proposed for African antelope by Hart et al. [18]. This model has come to be known as the programmed grooming model. Subsequently, Ben Hart, Lynette Hart, myself, and other colleagues (e.g., Andrew McKenzie, Bill Samuel, and Zhongqiu Li) published a total of 26 papers (not including review articles) that tested various aspects of the model. This effort has produced a robust body of work that provides strong confirmation of the programmed grooming model for ungulates, by which is meant the terrestrial hoofed mammals of the clade Euungulata (formerly Ungulata; [19]). In the 30+ years since the seminal 1992 paper, at least 10 studies have been published from unaffiliated investigators to test or evaluate the programmed grooming model in different host systems, including rodents and primates. Here, I review the work that has been carried out through 2023 and assess how the model has stood up over time. Although the programmed model has been applied to preening in birds, I will limit this assessment to mammals, for which the programmed grooming model was developed. Although the model was formulated with tick biology in mind, other ectoparasites (e.g., fleas, lice, and keds) will also be considered (Figure 4). My goal in this review is to evaluate to what extent the programmed grooming model can be applied to other taxa beyond ungulates and ticks and to discern the way forward for future research in this area.

**Figure 1 animals-14-01266-f001:**
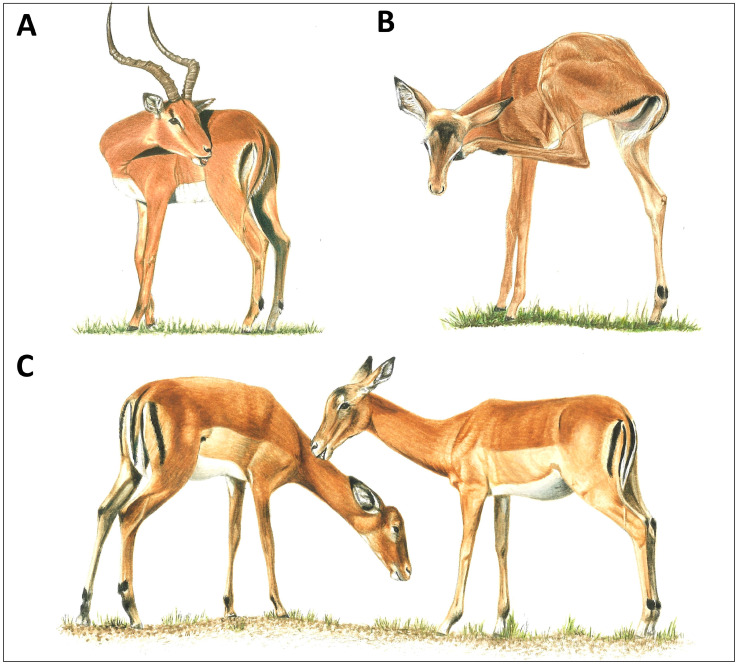
Grooming modes of a representative ungulate, the African impala (*Aepyceros melampus*). (**A**) Male oral self-grooming with teeth, (**B**) female scratching with hindleg hoof, and (**C**) females allogrooming (illustrations by Emma Mooring; (**A**,**B**) courtesy of [20]).

**Figure 2 animals-14-01266-f002:**
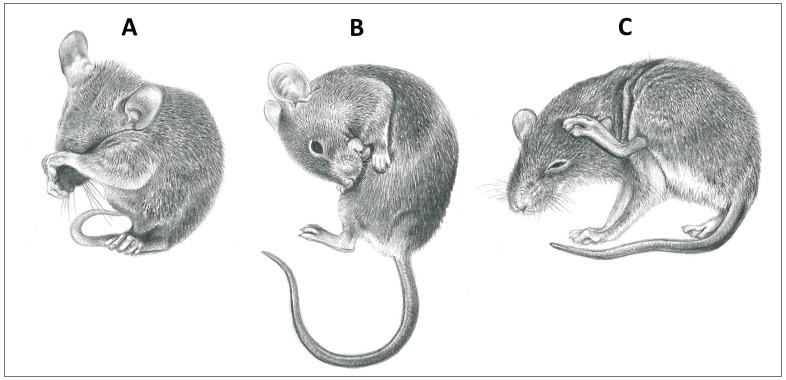
Grooming modes of mice (family *Muridae*). (**A**) Wipe with forelimbs, (**B**) oral self-grooming with tongue, and (**C**) scratch grooming with hindlimb (illustrations by Emma Mooring).

**Figure 3 animals-14-01266-f003:**
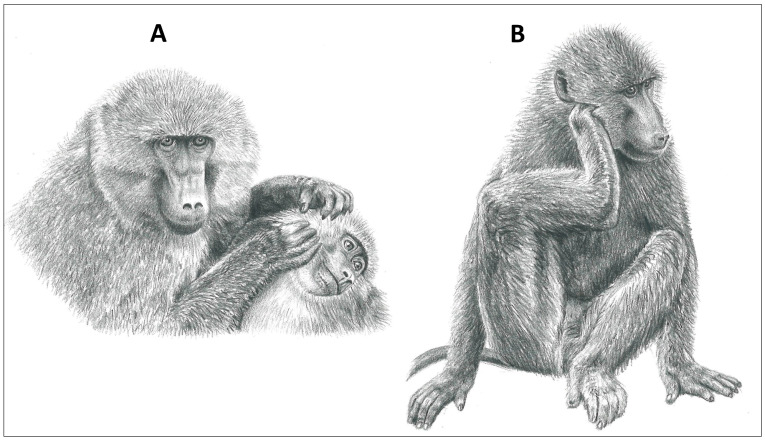
Grooming modes of a representative primate, the Chacma baboon (*Papio ursinus)*. (**A**) Allogrooming with hands and (**B**) scratch grooming with limb (illustrations by Emma Mooring).

**Figure 4 animals-14-01266-f004:**
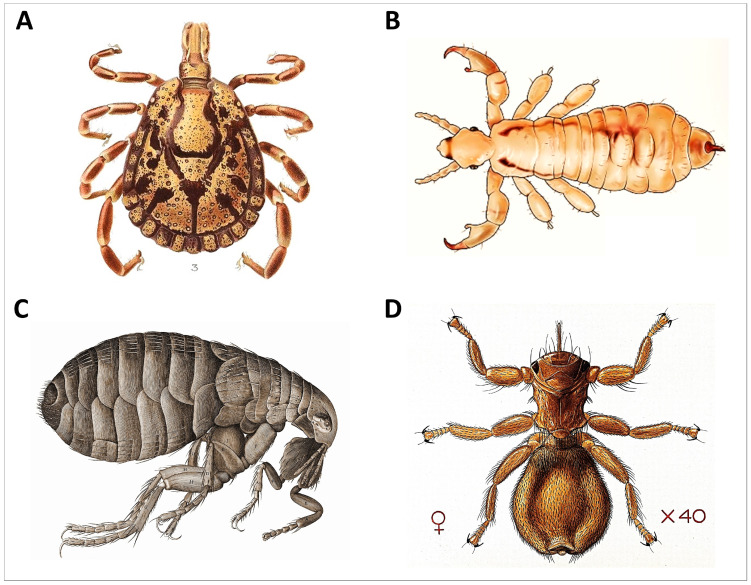
Representative ectoparasites of wildlife. (**A**) Hard tick, order *Ixodida*, (**B**) louse, order *Psocodea* (**C**) flea, order *Siphonaptera*, and (**D**) ked, order *Hippoboscidae* ((**A**): *Amblyomma marmoreum* male, Wilhelm Donitz, public domain via Wikimedia Commons; (**B**): head lice *Anoplura*, Clevelandclinic, CC BY-SA 4.0 via Wikimedia Commons; (**C**): flea *Siphonaptera*, Robert Hooke, public domain via Wikimedia Commons; and (**D**): sheep fly *Melophagus ovinus*, Creative Commons CC BY 4.0).

## 2. Review of the Programmed Grooming Model

Grooming behavior is the first line of defense against ectoparasite infestation for wild mammals. Animals with poor or restricted grooming behavior are vulnerable to excessive infestations from ticks and other ectoparasites (see above). Recent empirical studies have demonstrated the physiological costs of ectoparasitism in rodents, observing that the host condition improves when ectoparasites are removed [21]. For example, gerbils (*Gerbillus andersoni*) parasitized with high densities of fleas lost body mass due to higher energy requirements, with juveniles losing mass faster than the controls [21]. Similarly, juvenile jirds (*Meriones crassus*) exposed to fleas increased their grooming, but lost body mass in comparison with juveniles not exposed to fleas [22]. Turning to primates, a field study demonstrated that a heavy tick infestation in a troop of Chacma baboons (*Papio ursinus*) was responsible for over half of the infant mortality over 4 years [23].

Given the fitness costs of tick infestation, host animals have evolved morphological and behavioral adaptations to defend themselves against ectoparasite attacks. For example, the tongue is the primary grooming tool for cattle; indicine cattle (*Bos indicus*) are more resistant to tick infestation than taurine cattle (*B. taurus*) due to the greater density of filiform papillae that makes their tongue more effective in tick removal [24]. It has been argued that indicine cattle have evolved a rougher tongue under greater selection pressure from tick infestation in their tropical range [24]. Impala are antelope that have adapted to a tick-dense ecotone habitat, including a unique dental grooming apparatus that is reminiscent of the prosimian toothcomb, but with a loose movement of the incisors and canines, making a remarkably efficient tool for the removal of ticks from the pelage [16,25]. These morphological features have evolved to improve the effectiveness of grooming behaviors in the removal of ticks and other ectoparasites.

Two grooming models have been proposed to explain the endogenous (centrally regulated) and exogenous (peripherally activated) regulation of tick-defense grooming. The programmed grooming model postulates a type of central control (internal clock regulated by the CNS) that periodically evokes grooming behavior so as to remove ectoparasites like ticks before they attach and blood feed [18,20]. There is ample evidence for the central control of grooming [26,27,28,29,30]; also, see below. The stimulus-driven grooming model proposes that grooming is regulated as a response to host peripheral irritation from ectoparasite bites, which produce itching, burning, and pain via the release of ATP, serotonin, histamine, and bradykinin from injured cells, platelets, and mast cells [31,32,33,34]. The two models are not mutually exclusive and, indeed, are likely to operate concurrently as a complementary system [35].

### 2.1. Recent Advances in the Molecular Basis of Rodent Grooming

Grooming is an ancestral behavior common to both vertebrates and invertebrates [36,37]. Numerous studies demonstrate that grooming in mammals is an innate, preprogrammed behavior, consisting of fixed action patterns generated in the brainstem and specified by a genetic program [36]. In rodents, grooming accounts for up to 30–50% of daily activity and rodent grooming follows a complex sequence from head to tail that is evolutionarily conserved [38]. Significantly, grooming behavior is not dependent on tactile sensory feedback, but is, instead, modulated by basal ganglia in various parts of the brain [36]. Neural circuits within the forebrain are involved in controlling self-grooming behavior, along with the activity of the limbic region [39]. Various neurotransmitters modulate grooming behavior, with dopamine being the major excitatory neurotransmitter that amplifies self-grooming; in addition, glutamate has an excitatory effect and GABA has an inhibitory effect on grooming rate [39].

The ancient behavior of grooming is under the control of the Hox complex, specifically, the Hoxb8 transcription factor. Hoxb8 mutants experience a doubling in the rate of self-grooming and allogrooming [36]. The optogenetic stimulation of Hoxb8 microglia (macrophages) in specific areas of the brain (striatum or prefrontal cortex) induces grooming behavior through the activation of neurons and neural circuits [40]. Experiments suggest that Hoxb8 microglia function in opposition to non-Hoxb8 microglia to modulate grooming in mice, with Hoxb8 microglia acting as the brakes to downregulate grooming and non-Hoxb8 microglia accelerating grooming. In concert, these roles would allow for the fine-tuning of grooming rate [40].

### 2.2. General Predictions to Differentiate between the Models (Table 1)

When measures of grooming effort are associated with ectoparasite infestation, what type of correlation do the two models predict [18]? Because programmed grooming is preventive, this model predicts that individuals that groom the most will have the lowest density of ectoparasites because they have been groomed off. In contrast, the stimulus-driven model predicts that individuals that groom the most will harbor the highest density of parasites because this kind of grooming is positively correlated with the irritation caused by infestation. In other words, the stimulus-driven model predicts a positive correlation between grooming and ectoparasite infestation (more parasites → more grooming), while the programmed model predicts a negative correlation (more grooming → fewer parasites).

**Table 1 animals-14-01266-t001:** General predictions of the two grooming models.

Programmed Grooming Model	Stimulus-Driven Grooming Model
*Grooming is regulated by an endogenous mechanism independent of parasite bites*	*Grooming is regulated by exogenous peripheral stimulation from ectoparasite bites*
Grooming is preventive More grooming → fewer parasites	Grooming is reactive More parasites → more grooming
2. Grooming in absence of parasites Captive animals groom Grooming with few parasites	2. No grooming in absence of parasites No local immune response to trigger grooming behavior
3. Programmed grooming is modulated by local immune response to parasites Adjust baseline grooming rate to environmental parasite challenge Adjust baseline grooming rate to developmental body size changes	3. Changes in immune response to parasites modulates programmed grooming Grooming rate adjusts to immediate parasite stimulation from environment or body size changes during growth

### 2.3. Specific Predictions of Programmed Grooming (Table 2)

#### 2.3.1. Body Size Principle

The body size principle is based on the recognition that smaller animals, with a greater surface area-to-mass ratio, incur a higher physiological cost relative to larger animals, assuming a comparable density of infestation [18]. Because small animals have more surface area for parasites to attach to, but less blood volume for them to feed on, each parasite removes a larger proportion of the bodily resources of a smaller host. Thus, small-bodied animals should groom at a higher rate and should, consequently, maintain a lower density of ectoparasites compared with larger animals. The body size prediction can be applied interspecifically between different-sized species, or intraspecifically among age/sex classes of the same species. The intraspecific body size principle predicts that juveniles will groom more than adults and, for sexually dimorphic species, females will groom more than males. A corollary of the body size principle is that smaller individuals that groom more should carry a lighter load of ectoparasites than larger individuals that groom less. Juveniles of many ungulate species have been observed to groom more frequently than adults [41,42,43,44,45,46,47,48] and to harbor fewer ticks as a result [49].

**Table 2 animals-14-01266-t002:** Specific predictions of the programmed grooming model.

**1.** **Body size principle**	Cost of ectoparasites increases with decreasing body size due to greater surface-to-volume ratios in smaller animals *Interspecific Prediction*: smaller species will expend more effort in grooming (increased rate or efficiency) *Intraspecific Developmental Prediction*: juveniles will expend more grooming effort in grooming compared with adults *Intraspecific Sexual Dimorphism Prediction*: in dimorphic species, smaller females will expend more grooming effort compared with larger males
**2.** **Vigilance principle**	Cost of grooming exceeds fitness benefit for energy-limited breeding males that must prioritize vigilance for estrous females and rival males *Prediction 1*: breeding males will expend less grooming effort compared to non-breeding males or females *Prediction 2*: breeding males will carry a greater parasite load compared with non-breeding males or females
**3.** **Habitat principle**	Species have evolved a species-typical baseline grooming rate that matches the intensity of ectoparasite threat in their ancestral habitat *Prediction 1*: species that have evolved to inhabit a more parasite-dense habitat will expend greater grooming effort compared with species that have evolved in a habitat with less parasite threat *Prediction 2*: the first prediction will hold, even in a parasite-free or parasite-sparse environment, such as in captivity
**4.** **Tick challenge principle**	Species respond to short-term changes in parasite challenge by adjusting their grooming efforts *Seasonal Prediction*: because parasite challenge typically fluctuates according to season (wetter seasons support more parasites), individuals will expend greater grooming effort during the parasite-dense season *Programmed Prediction*: if grooming effort increases due to endogenous modulation of grooming, more grooming will result in fewer parasites *Stimulus-driven Prediction*: if grooming effort increases due to greater exogenous stimulation, more parasites will result in more grooming

#### 2.3.2. Vigilance Principle

The vigilance principle predicts that males of polygynous species will groom less than females in the same population during the breeding season so as to maintain high levels of vigilance for rival males or estrous females [18]. Testosterone has been shown to be the most likely mechanism driving this pattern, with higher levels of testosterone resulting in a physiological suppression of programmed grooming [35,45,50,51]. Sexually dimorphic grooming, in which females groom more than males, is, thus, the consequence of both the body size and the vigilance principle, with these principles acting independently and additively [35]. A corollary of the vigilance principle is that males that groom less should carry more ectoparasites than females. Sexually dimorphic grooming has been observed in a wide range of ungulates [13,14,18,44,45,46,48,50], with breeding male ungulates carrying many more ticks than females [13,14,52,53].

#### 2.3.3. Habitat Principle

The habitat principle recognizes that habitats with a greater density of ticks and other ectoparasites expose hosts to a higher risk of infestation and, thus, this principle predicts that animals that inhabit such areas should groom more frequently than those utilizing habitats of lower tick density. A broad generalization is that closed habitats, such as woodland and forest, have a greater abundance of ticks than open ones, such as grassland or savannah [54,55,56]. The programmed grooming model predicts that hosts adapted to infested habitats (e.g., woodlands) will groom at a higher rate than hosts adapted to low tick density (e.g., grassland), even when inhabiting a parasite-free environment due to evolutionary inertia.

A phylogenetic analysis in ungulates supported this prediction insofar as the evolution of complex oral grooming and adult allogrooming was concentrated in species inhabiting closed habitats versus open habitats, implying that lineages historically exposed to high levels of tick challenge in their ancestral environments tended to evolve grooming techniques that are more effective in removing ticks [35]. Thus, there is the tendency for ‘ticky’ habitats to favor the evolution of effective anti-parasite grooming behavior in ungulates [35].

#### 2.3.4. Tick Challenge Principle

The tick challenge principle predicts that grooming rate will broadly track the intensity of ectoparasite exposure [20]. Because parasite challenge may vary dramatically over time and space, and grooming behavior has costs, animals should adjust their grooming rate on a seasonal or geographical basis. The tick challenge prediction can support either the programmed or stimulus-driven model, depending on the general predictions (see above). The stimulus-driven model predicts that a greater infestation will stimulate a higher rate of grooming; thus, individuals that groom more will be more infested. In contrast, the programmed grooming model predicts that individuals that groom more will be less infested because ectoparasites have been preventively removed. Seasonally, impala in Zimbabwe self-groomed at the highest rate during the warm–wet season (January–April), when adult tick challenge was greatest, while grooming was reduced >40% during the hot–dry season (September–December), when adult tick abundance was at a minimum [20]. Because ticks were measured in the environment and not on individual impala [20], this study can support either the programmed or the stimulus-driven model. Geographically, the tick challenge principle has been termed the habitat principle.

### 2.4. Predictions of the Programmed Grooming Model

Support for programmed grooming can, thus, be obtained when any of the following predictions are tested and supported:Between species, individuals of smaller species will groom more frequently than those of larger species (interspecific body size principle). Usually, females are compared to control for intraspecific variation.Within a species, smaller juveniles will groom more frequently than larger adults and, for sexually dimorphic species, smaller females will groom more frequently than larger males (intraspecific body size prediction).Within a species, the grooming rate of actively breeding males will be lower compared with non-breeding males or adult females (vigilance prediction).Individuals exposed to a higher density of ectoparasite infestation will groom more than individuals exposed to lower infestation and such individuals will carry a lower ectoparasite load as a result (tick challenge principle).Species that are adapted to ectoparasite-dense environments will groom at a higher rate compared with species adapted to ectoparasite-sparse habitats, even in an environment with little or no ectoparasites (habitat principle).

### 2.5. Allogrooming

Allogrooming involves one individual grooming the body of another to remove debris or ectoparasites. While primates use their fingers to groom manually, other mammals use oral grooming methods such as licking, chewing, nibbling, and scraping to remove ectoparasites and comb through the hair [57]. Baboons that received more allogrooming had, in turn, lower tick loads and a higher packed red cell volume or hematocrit, a general measure of health status [58]. Although allogrooming in primates likely evolved to remove parasites from inaccessible body regions, it has often taken on social functions such as conflict reconciliation, tension relief, and bond formation [57], and this may be the case in other taxa. Allogrooming is associated with physiological changes, such as an increase in endorphins and neuropeptides, indicative of pleasure; reduced glucocorticoids, indicative of tension reduction; and reduced heart rates, suggestive of positive welfare [57]. In ungulates, phylogenetic evidence supports the evolution of allogrooming for parasite defense, with higher grooming rates concentrated in taxa with greater ectoparasite exposure [35]. The concentration of allogrooming on regions that are inaccessible to self-grooming is consistent with a utilitarian, parasite-removal function for allogrooming, such as with mule deer [59]. In rodents, a study of allogrooming in herb-field mice suggested a social function involving preparation for mating; nevertheless, allogrooming in this species appears to be regulated by an internal clock that supports the programmed grooming model [60].

Allogrooming in impala, an African antelope, is associated with a reciprocal pattern in which each animal alternates in delivering a bout of grooming to the head, neck, or shoulders of its partner, thus reducing the opportunity to ‘cheat’ by failing to reciprocate, as predicted by the generous tit-for-tat model of reciprocal altruism [61]. The fact that partner selection is not influenced by dominance or kinship, but by proximity, suggests that impala reciprocal allogrooming benefits the participants by providing hygienic benefits, i.e., the removal of ticks [41,62,63]. The allogrooming rates of juvenile impala are twice that of adults, supporting the body size prediction of programmed grooming [42,64]. In an experimental study with impala, the removal of ticks from control subjects resulted in a reduction in the rate of allogrooming, again supporting a tick-control function [9]. Hodgson et al. [57] has suggested that ungulates are a promising taxa for comparative allogrooming research, to explore the social and ecological factors impacting the evolution of sociality.

## 3. Results

### 3.1. Testing the Model Predictions

It is important to differentiate between tests of the model predictions and tests of the model itself. Predictions can be augmented or modified, but a poor model (i.e., hypothesis) can only be discarded. Only a few investigators have explored programmed grooming in other host or parasite systems and, in all cases, they have explicitly tested the predictions of the model. The following is a summary and assessment of these studies.

#### 3.1.1. Hawlena et al., 2008

Hawlena et al., 2008 [65] studied the grooming behavior of a desert rodent, Sundevall’s jird, to test the programmed grooming model. Their results did not support the body size principle insofar as the grooming rate of juvenile jirds in response to flea infestation was similar to and not significantly different from that of adult jirds. Both age classes responded to increased parasitism with increased grooming frequency, supporting the stimulus-driven model. The authors offer several plausible reasons for non-support of the body size principle in jirds. First, juvenile and adult rodents have less of a difference in surface-to-volume ratio (138%) compared with ungulates, and grooming frequency differences between juvenile and adult ungulates disappear by the time the surface-to-volume ratio declines to 150%. Therefore, the surface-to-volume differences between juvenile and adult rodents may not be sufficient to trigger grooming rate differences. Second, rodents are altricial and, therefore, juveniles may not be capable of fully functional grooming until their body size is similar to that of adults. Third, juvenile rodents have less dense fur than adults and, thus, less effective grooming by juveniles may have the same parasite-removal efficiency as adults. Finally, rodents are parasitized primarily by fleas, which are highly mobile, while ungulates are parasitized primarily by ticks, which are more sedentary. It is plausible that grooming is less effective in controlling fleas compared with ticks.

#### 3.1.2. Sarasa et al., 2011

Sarasa et al., 2011 [66] claimed to have experimentally tested and refuted the intraspecific body size prediction in Iberian ibex (*Capra pyrenaica*). However, the study was hampered by the absence of ticks or other ectoparasites on the animals. Instead, the investigators used what they called “pseudoectoparasites” (PEPs), waxed wooden triangles intended to mimic ectoparasites without triggering the host immune system. Given that both stimulus-driven and programmed grooming regulation depends on immunological stimulation and the fact that inanimate markers pose no biological cost to the host, there is no reason to expect that animals will respond to PEPs with programmed grooming patterns. The only behavioral observations in the study examined all grooming behaviors (self-oral grooming, allogrooming, and hindleg scratching combined) from scan sampling (scan intervals were not specified), with the results showing that young ibex (less than a year old) had the lowest percentage of all grooming scans as a percentage of their total activity budget, i.e., [all grooming scans/all scans] × 100. This unusual measure washed out the role of self-oral grooming (the grooming measure used in previous studies and most strongly associated with programmed grooming) by combining it with scratching (previously shown to not be under programmed control) and allogrooming (which has a social component) and then calculating the percentage of these scans out of all activity scans. Given that the activity budget of nursing juveniles <1 year of age is quite different from that of adult animals, it is impossible to draw any conclusions regarding the actual oral grooming rate of juvenile versus adult ibex. Had the investigators reported on the rate or duration of oral grooming in young juveniles versus adults, it is likely that the results would have supported the body size prediction.

#### 3.1.3. Heine et al., 2016

Heine et al., 2016 [67] tested the body size prediction in white-tailed deer (*Odocoileus virginianus*) and found support for the stimulus-driven model, because they found that fawns groomed at a higher rate than adults, but had a higher density of ectoparasites, ticks, and deer keds (i.e., positive correlation). They found that keds were a significant influence on grooming and they speculated that keds produced a higher level of irritation because they are mobile and can bite the host multiple times.

#### 3.1.4. Blank 2023

Blank 2023 [68] conducted field observations of goitered gazelle (*Gazella subgutturosa*) in Kazakhstan from 1981 to 1987, using continuous focal animal recording (mean observation time 3.4 h) and later transcribing his written records to the methodology used in the Hart–Mooring studies. Ectoparasites were not measured on the animals, but were assumed to be equal among age–sex classes based on a 1983 Russian language publication not available to the scientific community. Blank tested predictions of the programmed grooming model (the intraspecific body size principle and vigilance principle), neither of which were supported. These results are complicated by several factors. First, goitered gazelles observed in a tick-free zoological park [48] did support the body size prediction, with juveniles grooming at a much higher rate than adults. Second, ectoparasite load was not measured on the gazelles. Blank concluded that keds had the strongest impact on grooming rate and were much more numerous than ticks. Keds are bloodsucking, wingless insects that live permanently on their hosts. They differ from ticks in being fast-moving, able to feed multiple times with a painful bite, and they blood feed quickly, all of which suggests that grooming is less likely to be effective in controlling keds compared with ticks. Thus, there is conflicting evidence regarding this species; but, if goitered gazelle do not support the predictions of programmed grooming, they, and moose (*Alces alces*) [69], would be the only exceptions to the general trend in ungulates that have been studied.

#### 3.1.5. Summary

These studies indicate that the predictions of the programmed grooming model, which have been supported for the vast majority of ungulate species, were not supported in the rodents that were studied. Note that domestic cats (*Felis catus*) did support the programmed grooming model [8,70]. Many of these studies employed different data collection methods from the standard methodology employed by the Hart–Mooring studies, suggesting that, in some cases, the measures used might not have been appropriate for comparing grooming effort (e.g., [66]). More importantly, the evidence from these studies suggests that the failure to support the programmed grooming model is not necessarily due to a flaw in the model hypothesis itself, but may be the outcome of inadequate predictions of the model. It should be borne in mind that the predictions of the programmed grooming model (especially in regard to body size and sexual dimorphic grooming) were formulated based on the biology of ungulate hosts—e.g., antelope, deer, sheep, and goats—and tick ectoparasites, rather than the very different biology of rodent hosts or non-tick ectoparasites (e.g., keds, fleas, or inanimate markers). The predictions that operate for tick-defense grooming by most ungulates may not be appropriate for other taxa of hosts or parasites. Differences in host grooming behavior or ectoparasite biology may require different predictions to support programmed grooming. For example, Malange et al. [37], in a review of the evolution of grooming in rodents, suggested that a reduced body size may select for more efficient grooming patterns in addition to a greater frequency of grooming. Perhaps the dorsoventral sequence of cephalo-caudal (head-to-tail) grooming observed in sciurognathids (rodents) would prevent crossed-infestations between body parts and, thus, be more efficient in removing parasites. Whereas ungulates utilize a standardized oral grooming technique using teeth, tongue, or both, rodents and other taxa may exhibit greater variability in grooming methods, some of which may be more efficient than others in removing parasites.

Finally, it is also possible that ungulates have evolved programmed grooming in response to their unique ecological niche and life history strategy, which might not fully apply to other taxa. Some examples of different evolutionary trajectories in grooming behavior come to mind. For example, hosts living in a low-parasite (or no-parasite) habitat may never have had the need to evolve tick-defense grooming behavior. This might include Pere David’s deer (*Elaphurus davidianus*), maintained in captivity for hundreds of years [71]; Chihuahuan desert bighorn sheep, living in a tick-free habitat [45]; or moose that recently arrived in North America and lack an evolutionary history with the winter tick [69]. Although unstudied, Arctic muskox (*Ovibos moschatus*) and Alaskan caribou (*Rangifer tarandus*) have lived in tick-free environments for millennia, relaxing the selection for parasite-defense behaviors such as grooming. Gregarious social species such as impala, bison (*Bison bison*), or elk (*Cervus elaphus*) may be subject to different ectoparasite selection pressures compared with solitary species like moose, which do not exhibit the grooming patterns predicted by the programmed grooming model [69]. Hosts from different phylogenetic lineages are likely to have evolved different defensive behaviors against parasites, as illustrated by ungulates versus rodents [37]. Interestingly, a study by Cooper et al. [72] suggested that greater parasite burdens were linked to higher host mortality in ungulates, but not in carnivores or primates. This is just one piece of the puzzle, but suggests that ungulates may respond to parasites differently from other mammalian taxa.

### 3.2. Testing the Model Itself

Even if the predictions of the programmed grooming model are not completely appropriate for other taxa of hosts or for other parasites, can the model hypothesis itself be supported? There are at least two ways to support programmed grooming when observations of grooming behavior and ectoparasite load data are available for individual hosts.

If individuals that groom more have fewer parasites (i.e., grooming rate is negatively correlated with parasite load or density). Note that stimulus-driven grooming predicts the opposite, that individuals with more parasites will groom more.If individuals maintain a baseline rate of grooming in a parasite-free or parasite-sparse environment. Even better, if individuals display the predicted differences in body size, sex, or breeding status in a parasite-free/sparse environment.Note that these criteria were formulated with tick parasitism in mind. Would these criteria be different when considering other parasites, such as fleas or keds?

I now address studies in which these parameters can be assessed to test the model itself.

#### 3.2.1. Stopka and Graciasova 2001

Stopka and Graciasova 2001 [60] found that herb-field mice (*Apodemus uralensis*) engage in both self-grooming and allogrooming and that grooming operates independently of parasite exposure. They found that self-grooming in the herb-field mouse is a stochastic process, where each bout is unpredictable but regularly performed in the absence of ectoparasites such as ticks and fleas. These results support an endogenous regulatory grooming mechanism, i.e., they are in agreement with the programmed grooming model.

#### 3.2.2. Yamada and Urabe 2007

Yamada and Urabe 2007 [73] observed the grooming behavior of sika deer (*Cervus nippon*) in two populations, one in which tick density was high and the other in which tick density was low. The frequency of grooming by deer in the high-tick population fluctuated with tick density, whereas the self-grooming by deer in the low-tick population did not correlate with tick density. However, the overall duration of grooming did not differ between the two populations, implying the programmed grooming model.

#### 3.2.3. Akinyi et al., 2013

Akinyi et al., 2013 [58] supported programmed grooming in yellow baboons (*Papio cynocephalus*) insofar as baboons that received more allogrooming in the six months prior to tick removal had lower tick loads, while baboons with lower tick loads had a higher packed red cell volume (or hematocrit), a measure of health (low hematocrit would indicate anemia).

#### 3.2.4. Eads et al., 2017

Eads et al., 2017 [74] studied the grooming behavior of black-tailed prairie dogs (*Cynomys ludovicianus*) in colonies that were dusted with a pulicide to kill fleas (parasite-free) versus those in a colony that was not dusted (parasite-present). They found that, although the prairie dogs in the undusted colony groomed at a higher rate than those in the dusted colony (supporting the stimulus-driven grooming hypothesis), those prairie dogs in the dusted colony, with no fleas or ectoparasites on their bodies, still performed a significant level of grooming, which supports the programmed grooming model.

#### 3.2.5. Rayner et al., 2022

Rayner et al., 2022 [75] discussed two examples of ‘vestigial behaviors’, in which grooming behavior persisted after a long period of time with little or no parasite exposure. These examples support the programmed grooming model insofar as the stimulus-driven model would predict that rates of grooming would drop to zero in the absence of tick exposure. Desert bighorn sheep (*Ovis canadensis mexicana*) still exhibited tick-defense grooming behaviors, including body size differences predicted by programmed grooming, even though ticks have been absent from the Chihuahuan Desert for thousands of years [46]. Similarly, Pere David’s deer still exhibit the body size principle predicted by programmed grooming despite hundreds of years in captive environments in which tick exposure was either absent or very low [71].

It should be mentioned that moose are a counter-example insofar as they do not demonstrate any of the patterns of programmed grooming, but rather groom in a manner that supports stimulus-driven grooming [69]. Moose that had a higher density of ticks groomed more, as shown in the following two ways: (1) moose calves had the highest density of tick load and also groomed at the highest rate, and (2) moose dramatically increased grooming efforts during the late spring, when adult winter ticks produced much more irritation and removed orders of magnitude more blood than larval or nymphal ticks in the prior fall or winter. The preventive grooming of larval and nymphal ticks would be expected with the programmed grooming model, which may be missing from moose because of their more recent arrival in North America and their shorter exposure to the winter tick.

## 4. Conclusions and a Personal Reflection

### 4.1. History of the Programmed Grooming Model

When Ben and Lynette Hart first started testing their new hypothesis about tick-defense grooming, it was anyone’s guess how well the model predictions would be supported. But the initial results with African antelope—Thomson’s gazelle (*Gazella thomsonii*), Grant’s gazelle (*G. granti*), wildebeest (*Connocheates gnu*), and impala—fit the model very well and it appeared that they were on to something [18]. At the time, I was in the UC Davis Animal Behavior Graduate Group carrying out my Ph.D. under the supervision of Ben; my dissertation subject was impala. I subsequently, spent years meticulously documenting the behavior and ecology of impala, first at the San Diego Wild Animal Park and then in Zimbabwe, South Africa, and Namibia. For my first postdoc, we collaborated with Andrew McKenzie to conduct an experimental study of impala grooming and tick infestation. All the impala data perfectly supported the model predictions; therefore, for my second postdoc at the University of Alberta under the supervision of Bill Samuel, I applied the same approach to moose, elk, and bison. This entire body of research utilized the same methodology and metrics of grooming across multiple species and continents. Indeed, my trusty Tandy 102 portable computer was used to code behavioral observations throughout my dissertation and postdoctoral research and into my early academic career.

The results for elk and bison agreed with the antelope data [43,44], but moose did not [69]. However, because moose are devastated by several parasites that do not seriously impact odocoiline deer or elk (meningeal worm, liver flukes, and winter ticks), it seemed plausible that the apparent failure of moose to evolve programmed grooming patterns could be attributed to the relatively recent exposure of moose to North American parasites compared to other cervids. With my undergraduate students, we took the research to the next level with comparative studies of captive ungulates at a tick-free zoological park [35,48,76], all of which provided robust support for the predictions of programmed grooming based on 36–60 ungulate taxa. Single-species field observations of desert bighorn sheep and bison provided additional support and clarification, while collaboration with Zhongqiu Li and other Chinese colleagues offered yet more support in Pere David’s deer, Tibetan antelope (*Pantholops hodgsonii*), and Tibetan gazelle (*Procapra picticaudata*). Although moose, and possibly goitered gazelle, apparently do not support the model, these are outliers to the broad support for the programmed grooming model among the hoofed mammals.

### 4.2. Conclusions

The studies presented here, especially those that attempted to test the programmed grooming model predictions in non-ungulate systems, have provided some new insights. The best insights come out of investigations of grooming in the order Rodentia (e.g., [37,65]).

The Hart–Mooring studies used a standardized methodology that did not vary across dozens of field investigations. However, the new studies generally employ different data collection methods, which often make it difficult or impossible (e.g., [66]) to compare studies or to interpret the results in light of the programmed grooming predictions. In my opinion, any investigation seeking to explicitly test the programmed grooming model must make every effort to use the same methods and metrics that we used, if this is possible.The predictions of the programmed grooming model should be tailored to the biology of the host under investigation. The original formulation of the model predictions was based on ungulate hosts, which may not be appropriate for alternative hosts. For rodents, in which altricial juveniles may have a lighter hair coat, less surface-to-volume ratio differences compared with adults, and are incapable of fully functional self-grooming, testing the body size principle by comparing the grooming rate of juveniles versus adults may not be appropriate. It might be more useful to predict that juveniles will employ more efficient grooming patterns, such as the dorsoventral sequence of grooming observed in sciurognathid rodents [37].Similarly, the predictions of the programmed grooming model should be tailored to the biology of the ectoparasite involved. Because the model predictions were based on tick ectoparasites, some aspects of the model may be inappropriate for alternative parasites, such as lice, fleas, or keds. Unlike ticks, which move slowly, take a long time to blood feed, feed only once, and produce only slight cutaneous irritation, fleas and keds are highly mobile, feed multiple times, and ked bites are rather painful. One might expect that the effectiveness of a programmed grooming system could be different for these ectoparasites compared with ticks and may, thus, require new predictions.

In conclusion, moving forward in our understanding of the biological factors of parasite-defense grooming behavior will require both old and new approaches. The tried-and-true standard methodology and behavioral measures used in previous studies should be retained, when possible, to allow for meaningful comparisons. On the other hand, the predictions arising from a programmed grooming system must, out of necessity, be different for new hosts and parasites—this will take new approaches and a full understanding of the ecology, evolution, molecular biology, and ethology of host–parasite grooming systems in nature. There is still much to learn about the role of grooming in enabling wild animals to stay healthy despite the challenge of parasites.

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
