# Peer review of "Programmed Grooming after 30 Years of Study: A Review of Evidence and Future Prospects"

_animals, 2024, doi:10.3390/ani14091266_

Round 1

Reviewer 1 Report

Comments and Suggestions for Authors

The submitted manuscript is a review whose main aim is to present a review of the studies that have been published during the last 30 years to test or evaluate the programmed grooming model to control ticks and other ectoparasites in different host systems, primarily in wild ungulates and other taxa, including rodents and primates.

 Overall, the manuscript is well written, presents an updated state of the art on the subject by summarizing the literature published on the use of the evolutionary model for the endogenous regulation of tick-removal grooming in wild ungulates.

 The manuscript is divided in sections with a general introduction, a review of the Programmed Grooming Model, including a detailed comparisson with the Stimulus-driven Grooming Model, a results section describing studies performed in other hosts or parasite systems, and a conclusions section, providing insights into the understanding of the biological factors of the parasite-defense function of grooming behavior, for which both old and new approaches will required for new hosts and parasites – as well as a full understanding of the ecology, evolution, molecular biology, and ethology of the grooming systems in nature

Minor details authors need to take care off are as follows:

Line 22. Delete (222)

Line 39. Delete (243)

Line 53. Change (Little 1963; Sutherst et al. 1983; Kaiser et al. 1991) for [1,2,3]

 Please refer to the Animals’ instructions to authors to see how to best cite the references (numbers in brackets) once they first appear in the manuscript

 For example: In the text, reference numbers should be placed in square brackets [ ], and placed before the punctuation; [1], [1–3] or [1,3]. The reference list should include the full title, as recommended by the ACS style guide. References should be described as follows, depending on the type of work:

 “Journal Articles:

1. Author 1, A.B.; Author 2, C.D. Title of the article. Abbreviated Journal Name YearVolume, page range.

Books and Book Chapters:

2. Author 1, A.; Author 2, B. Book Title, 3rd ed.; Publisher: Publisher Location, Country, Year; pp. 154–196.
3. Author 1, A.; Author 2, B. Title of the chapter. In Book Title, 2nd ed.; Editor 1, A., Editor 2, B., Eds.; Publisher: Publisher Location, Country, Year; Volume 3, pp. 154–196.”

Author Response

REVIEWER #1

Minor details authors need to take care off are as follows:

Line 22. Delete (222) The word count for the Simple Summary was deleted.

Line 39. Delete (243) The word count for the Abstract was deleted.

Line 53. Change (Little 1963; Sutherst et al. 1983; Kaiser et al. 1991) for[1,2,3]. Please refer to the Animals’ instructions to authors to see how to best cite the references (numbers in brackets) once they first appear in the manuscript. For example: In the text, reference numbers should be placed in square brackets [ ], and placed before the punctuation; [1], [1–3] or [1,3]. The reference list should include the full title, as recommended by the ACS style guide. References should be described as follows, depending on the type of work: [followed by an excerpt from the style guide on how to cite journal articles, etc.]

The reviewer requested that the in-text citations and literature cited section conform to the MDPI style. I had left these to do once all the other revisions were made, since any other changes can change all of the reference numbers under this citation system. The citations now conform to the journal style.

Reviewer 2 Report

Comments and Suggestions for Authors

The paper explores the programmed grooming model, focusing on its historical development, empirical testing across various mammalian taxa, and implications for understanding grooming behavior in response to ectoparasites. Through a review of studies, the paper tests the model's precision in some species while acknowledging the need for tailored predictions based on host and parasite biology. The paper appears to be of high quality. Overall, it seems to be well-researched, well-written, and informative.

The following minor suggestions are proposed for the authors' consideration to enhance clarity:

Consider revising the title to "Programmed Grooming After 30 Years of Study: A Review of Evidence and Future Prospects "

Line 10. Clarify "central control" in order to help readers better understand the mechanism being discussed

Line 38. The phrase "There is still much to learn" could be perceived as slightly colloquial. It can be rephrased for a more formal tone, such as "Further research is warranted to enhance our understanding of the role of grooming in maintaining the health of animals facing parasite challenges."

Line 46. Consider revising "broadly defined" to "broadly speaking" for a slightly more formal tone.

Line 62. Replace "individuals with effective grooming behavior" with "individuals exhibiting effective grooming behavior" for clearer phrasing.

Line 145. Replace "states" with "suggests" for a more cautious interpretation of the grooming model.

Line 157. Instead of "complex cephalo-caudal sequence," consider "complex sequence from head to tail" for clarity.

Line 382-383. "which might not fully apply to other taxa.": Specify the potential limitations.

Author Response

REVIEWER #2

The following minor suggestions are proposed for the authors' consideration to enhance clarity:

  1. Consider revising the title to "Programmed Grooming After 30 Years of Study: A Review of Evidence and Future Prospects " The title has been changed as suggested, as it is clearer.
  2. Line 10. Clarify "central control" in order to help readers better understand the mechanism being discussed. I agree with the reviewer and have made changes in the main text at first mention of “central control” (lines 134-135) to clarify that by central control I mean an “internal clock regulated by the CNS…”
  3. Line 38. The phrase "There is still much to learn" could be perceived as slightly colloquial. It can be rephrased for a more formal tone, such as "Further research is warranted to enhance our understanding of the role of grooming in maintaining the health of animals facing parasite challenges." I have made the suggested change to the conclusion.
  4. Line 46. Consider revising "broadly defined" to "broadly speaking" for a slightly more formal tone. I have made the suggested change.
  5. Line 62. Replace "individuals with effective grooming behavior" with "individuals exhibiting effective grooming behavior" for clearer phrasing.    I have made this change.
  6. Line 145. Replace "states" with "suggests" for a more cautious interpretation of the grooming model. This has been done.
  7. Line 157. Instead of "complex cephalo-caudal sequence," consider "complex sequence from head to tail" for clarity. The suggested change was made.
  8. Line 382-383. "which might not fully apply to other taxa.": Specify the potential limitations.

This was an excellent suggestion and therefore I have added the following paragraph (lines 397-416):

Finally, it is also possible that ungulates have evolved programmed grooming in response to their unique ecological niche and life history strategy, which might not fully apply to other taxa. Some examples of different evolutionary trajectories in grooming behavior come to mind. For example, hosts living in a low-parasite (or no-parasite) habitat may never have had the need to evolve tick-defense grooming behavior. This might include Pere David’s deer (Elaphurus davidianus) maintained in captivity for hundreds of years [72], Chihuahuan desert bighorn sheep living in a tick-free habitat [45], or moose that recently arrived in North America and lacking an evolutionary history with the winter tick [70]. Although unstudied, Arctic muskox (Ovibos moschatus) and Alaskan caribou (Rangifer tarandus) have lived in tick-free environments for millennia, relaxing the selection for parasite-defense behavior such as grooming. Gregarious social species such as impala, bison (Bison bison), or elk (Cervus elaphus) may be subject to different ectoparasite selection pressures compared with solitary species like moose, which do not exhibit the grooming patterns predicted by the programmed grooming model [70]. Hosts from different phylogenetic lineages are likely to have evolved different defensive behaviors against parasites, as illustrated by ungulates versus rodents [37]. Interestingly, a study by Cooper et al. [73] suggested that greater parasite burdens were linked to higher host mortality in ungulates, but not in carnivores or primates. This is just one piece of the puzzle, but suggests that ungulates may respond to parasites differently from other mammalian taxa.

Reviewer 3 Report

Comments and Suggestions for Authors

Comments on the manuscript “Programmed grooming at 30: a tricennial review of the evidence and future prospects” submitted to the Animals

 General comments

The manuscript is a critical review of Programmed grooming (PG), a hypothesis by Lynette and Ben Hart in the early 90’s. The author presents a series of evidence that reinforces the hypothesis but also shows articles in which the hypothesis is rejected. The author compares the Programmed grooming hypothesis with the stimulus-driven (MSD) model, highlighting that both are not mutually exclusive. While Programmed grooming appears to be preventive, the stimulus-driven model is committed, to the effective removal of ectoparasites as much as possible.

The article is enlightening and makes an honest critique of the limitations of the PG hypothesis. Despite the arguments for and against, I found it difficult to connect some articles that exemplify the arguments, such as the article by Blank (2023) which has many flaws in applying the model.

Surprisingly, the PG model ignores the evolutionary and functional strength of allogrooming, which is well established in some species to increase cohesion and reinforce social hierarchy (e.g., Lazaro-Perea et al., 2004). Another flaw in the model is the lack of work on characteristically solitary ungulates, such as Blastocerus dichotomus, Tapirus terrestris, and some of the genus Mazama. There are many difficulties in studying these species in the wild, but it would be intriguing to test the hypotheses in these solitary species.

The manuscript, with this collection of 30 years of knowledge generated regarding the GP and MSD hypothesis, is perhaps a strong provocation for researchers to test on other species and with methodologies that are similar or identical to what the author proposes.

One of the highlights of this article is that it was written by an author whose graduate studies were coordinated by Lynette and Ben Hart, later became a very active contributor. This collaboration between the trio of authors and other collaborators supposedly generated many discussions and knowledge that will still emerge in the coming years, both in the group under the author's leadership and in other labs worldwide.

I believe the article has a great impact on reviving the hypothesis and clarifying strengths and weaknesses, which will stimulate further studies.

I see all grooming and allogrooming findings as being related to parasitic infestation, an open system. The hypothesis, in this case, as the author makes it clear, seems to have more limitations than generalizations. To be seen in the future, we hope.

REFERENCE

Lazaro-Perea, C., de Fátima Arruda, M., & Snowdon, C. T. (2004). Grooming as a reward? Social function of grooming between females in cooperatively breeding marmosets. Animal Behaviour, 67(4), 627-636.

Author Response

REVIEWER #3

  1. Surprisingly, the PG model ignores the evolutionary and functional strength of allogrooming, which is well established in some species to increase cohesion and reinforce social hierarchy (e.g., Lazaro-Perea et al., 2004).

I originally had not included allogrooming because all of the new work was focused on self-grooming, but I agree with the reviewer that a discussion of allogrooming should be included, especially regarding the role allogrooming may play in a parasite-removal function. I have therefore added a new section of 2 paragraphs (lines 264-296):

2.5. Allogrooming

Allogrooming involves one individual grooming the body of another to remove debris or ectoparasites. While primates use their fingers to groom manually, other mammals use oral grooming methods such as licking, chewing, nibbling, and scraping to remove ectoparasites and comb through the hair [57]. males. Baboons that received more allogrooming, had in turn lower tick loads and higher packed red cell volume or haematocrit, a general measure of health status [58]. Although allogrooming in primates likely evolved to remove parasites from inaccessible body regions, it has often taken on social functions such as conflict reconciliation, tension relief, and bond formation [57], and this may be the case in other taxa. Allogrooming is associated with physiological changes, such as an increase in endorphins and neuropeptides indicative of pleasure, reduced glucocorticoids indicative of tension reduction, and reduced heart rates suggestive of positive welfare [57]. In ungulates, phylogenetic evidence supports the evolution of allogrooming for parasite defense, with higher grooming rates concentrated in taxa with greater ectoparasite exposure [35]. The concentration of allogrooming on regions that are inaccessible to self-grooming is consistent with a utilitarian, parasite-removal function for allogrooming, such as with mule deer [59]. In rodents, a study of allogrooming in herb-field mice suggested a social function involving preparation for mating; nevertheless, allogrooming in this species appears to be regulated by an internal clock that supports the programmed grooming model [60].

Allogrooming in impala, an African antelope, is associated with a reciprocal pattern in which each animal alternates in delivering a bout of grooming to the head, neck, or shoulders of its partner, thus reducing the opportunity to ‘cheat’ by failing to reciprocate, as indicated by the generous tit-for-tat model of reciprocal altruism [61]. The fact that partner selection is not influenced by dominance or kinship, but by proximity, suggests that impala reciprocal allogrooming benefits the participants by providing hygienic benefits, i.e., the removal of ticks [41, 62-63]. Allogrooming rates of juvenile impala are twice that of adults, supporting the body size prediction of programmed grooming [64-65]. In an experimental study with impala, the removal of ticks from control subjects resulted in a reduction in the rate of allogrooming, again supporting a tick-control function [9]. Hodgson et al. [57] has suggested that ungulates are a promising taxa for comparative allogrooming research to explore the social and ecological factors impacting the evolution of sociality.

  1. Another flaw in the model is the lack of work on characteristically solitary ungulates, such as Blastocerus dichotomus [marsh deer], Tapirus terrestris [lowland tapir], and some of the genus Mazama [brocket deer]. There are many difficulties in studying these species in the wild, but it would be intriguing to test the hypotheses in these solitary species.

Because the programmed model has not been tested in these or other solitary species (apart from moose), I cannot include empirical studies or provide specific data. However, I have included this concept to the conclusions section mentioned above (lines 408-411): “Gregarious social species such as impala, bison (Bison bison), or elk (Cervus elaphus) may be subject to different ectoparasite selection pressures compared with solitary species like moose, which do not exhibit the grooming patterns predicted by the programmed grooming model [70].”